# The Community Health Supporting Environments and Residents’ Health and Well-Being: The Role of Health Literacy

**DOI:** 10.3390/ijerph18157769

**Published:** 2021-07-22

**Authors:** Tianfeng He, Lefan Liu, Jing Huang, Guoxing Li, Xinbiao Guo

**Affiliations:** 1Department of Occupational and Environmental Health Sciences, School of Public Health, Peking University, 38 Xueyuan Road, Beijing 100191, China; hetfnbcdc@163.com (T.H.); jing_huang@bjmu.edu.cn (J.H.); liguoxing@bjmu.edu.cn (G.L.); 2Ningbo Municipal Center for Disease Control and Prevention, Ningbo 315010, China; 3Center for Health Economics, School of Economics, University of Nottingham Ningbo China, Ningbo 315100, China; lefan.liu@nottingham.edu.cn

**Keywords:** health literacy, health supporting environment, green space, healthy city, healthy setting

## Abstract

We evaluate the impacts that health supporting environments have on residents’ health and well-being. Using a stratified multi-stage sampling method, we select a sample of 12,360 permanent adult residents aged 15–69, and collect information on their health literacy level, as well as their demographic background and health. This individual level data is then merged with the administrative health supporting environment data. More than two thirds of residents self-reported having good/excellent health, and the percent of adults living in communities with healthy parks, healthy trails, and healthy huts in their community is 23 percent, 43 percent, and 25 percent, respectively. Controlling for a series of confounding factors at the community and individual levels, we find that healthy parks and healthy trails are positively correlated with self-reported health, which increases the probability of self-reporting good health by 2.0 percentage points (*p* < 0.10) and 6.0 percentage points (*p* < 0.01), respectively. Access to healthy huts is negatively associated with self-reported health, decreasing the probability of self-reporting good health by 5.0 percentage points (*p* < 0.01). Health literacy plays a role in moderating the effect of health parks, and a positive effect is more likely to be observed among adults with lower health literacy. Health supporting environments may play a role in reducing the likelihood of undiagnosed diseases and changing residents’ lifestyles, which promotes the health and well-being of residents, especially among those with inadequate health literacy.

## 1. Introduction

As of 2018, 55% of the world’s population—4.2 billion people—live in urban areas [1]. This trend is expected to continue. By 2050, the urban population is expected to more than double its current size, and around 68% of the world’s population will live in cities [1]. Many issues in health and environment are likely to arise with rapid urbanization. To deal with these threats, WHO has begun to advocate the construction of healthy cities as a global strategy since the 1980s. The Healthy Cities strategy has been adopted as the national development strategy of many countries around the world. This strategy is also critical to the implementation of the UN’s New Urban Agenda [2] and the achievement of the UN Sustainable Development Goals (SDGs) [3]. A common theme in these documents is that a sustainable development depends increasingly on the successful management of urban growth, especially in low-income and lower-middle-income countries where the most rapid urbanization is expected between now and 2050 [1].

Over the past few decades, healthy cities have been successfully developed worldwide. According to WHO, a healthy city is one that is “continually creating and improving those physical and social environments and expanding those community resources which enable people to mutually support each other in performing all the functions of life and developing to their maximum potential” [4]. To achieve the goal of a healthy city, it is important to create a health supporting environment via healthy setting approaches. Settings for health or healthy settings, refer to “the place or social context in which people engage in daily activities in which environmental, organizational and personal factors interact to affect health and wellbeing” [4]. Today, various settings are used to facilitate the improvement of public health throughout the world and examples of settings include schools, work sites, hospitals, villages, and cities [5]. The importance of health supporting environments is also documented in the literature of the psychology of sustainability and sustainable development. The key idea is to place the psychological well-being of human beings in different environments, from the natural environment, the personal environment, the social environment, the organizational environment, and the inter-organizational environment, to the globalized environment and the virtual environment [6].

For China, the national healthy setting movement can be traced back to the National Healthy Lifestyle for All (NHLA) campaign in 2007. The NHLA campaign calls on “the Chinese people to develop a healthy diet and engage in physical exercises, advocating healthy lifestyle ideas, creating a supporting environment for a healthy lifestyle, and enhancing the people’s awareness and behavioral abilities to develop a healthy lifestyle” [7], which coincides with the UN’s SDGs on good health and well-being (i.e., ensure healthy lives and promote well-being for all at all ages). Following the NHLA missions, the Chinese government issued the “Health Supporting Environment Development Guideline for the National Healthy Lifestyle for All” in 2013 (hereinafter referred to as the “Guideline”) [8]. In this Guideline, governments at regional levels are required to build nine types of health supporting environments, including healthy worksites, healthy communities, health schools, health canteens, healthy restaurants, healthy trails, healthy huts, healthy street and healthy parks [8]. Up until 2018, over 60,000 health supporting environments were built and accredited nationwide [9]. From an empirical perspective, the growth may be considered as meeting the national guidelines. However, evaluating the quality of these environments remains an important task to provide guidance for future policies. Therefore, the primary goal of our paper is to study the relationship between health supporting environments and residents’ health and well-being. It is worth noting well-being—often synonymous with quality of life—is widely used in many different fields. In the field of urban spaces and architecture, quality of life generally refers to the condition of the environment in which people live and/or some attribute of people themselves, such as their health or educational achievement [10]. Here we use self-reported health status to indicate health-related quality of life, which is often viewed as a multidimensional concept that incorporates symptoms, functional ability, and perceptions of health in the field of health care [11,12]. Although our measures may not holistically convey many aspects of the notion of health-related quality of life, we believe our study is a starting point to provide a larger platform for multi-disciplinary dialogue between public health and urban spaces.

Health literacy (HL) is another topic with growing importance in the field of public health, which is particularly true in China. The first “National Health Literacy Surveillance” (NHLS) survey was conducted in 2008, in which the Chinese government adopted the widely accepted definition of health literacy, which was developed by the (US) National Library of Medicine—“the degree to which individuals can obtain, process, and understand the basic health information and services they need to make appropriate health-related decisions”. The first NHLS surveyed around 80,000 residents aged 15–69 and included 96 items intended to measure adult health literacy. The national health literacy rate in 2008 is merely 6.48%. This rate measures the percentage of residents who gave correct answers to at least 80% of the survey questions, i.e., classified as having adequate health literacy. A large rural–urban disparity exists with the rate of health literacy being 9.49% among urban residents and only 3.43% among rural residents [13]. The second NHLS survey was conducted in 2012 and the survey has run annually since then. This rate of health literacy rose steadily from 8.8% in 2012 to 10.25% in 2015. In 2016, the Chinese government issued its “Healthy China 2030 Blueprint”, which proposed major health indicators to be achieved in 2030. In this blueprint, the rate of national health literacy is aimed to increase to 30%, tripling the existing level in 2015. 

Although empirical evidence on the relationship between health literacy and health outcomes remains mixed, a growing body of literature has found that lower health literacy is associated with poorer health outcomes [14,15,16,17]. As a result, promoting health literacy is a public health goal in many countries. In an integrated framework that draws on several models of health literacy, health literacy plays a key role that determines an adult’s health outcomes at different stages of health behavior change [18]. Our secondary goal thus, is to explore the role that health literacy plays in mediating the impact that access to health supporting environments has on residents’ health and well-being. 

## 2. Materials and Methods

### 2.1. Data

To address our two research questions, we use data from Ningbo. Ningbo is the second largest city in Zhejiang province, which is one of the provinces in eastern China and is among the top five provinces in terms of the total number of health supporting environments [9]. Ningbo is also an important commercial and financial center in South China [19]. Located in the Yangtze River Delta in South China, Ningbo was ranked as the world’s fourth-largest port city in 2013.

Our data comes from 2018 and 2019 National Health Literacy Surveillance (NHLS) conducted in Ningbo. The population in Ningbo is 6.03 and 6.08 million in 2018 and 2019, respectively. This survey is representative of the institutionalized residents aged 15–69 years old who had lived in Ningbo for more than 6 of the previous 12 months, regardless of whether they had local household registration (i.e., hukou in Chinese). 

We used a stratified multi-stage PPS (probabilities proportional to population size) sampling frame. There are 10 districts (or counties) in Ningbo. At each county, we selected 4 communities (or townships), and then selected 2 neighborhoods (or villages) within each community (or township). All neighborhoods within each community represent urban regions and all villages within each township represents rural regions. If there were greater than 750 but less than 1500 households within a neighborhood (or village), the unit is regarded as a primary sampling unit (PSU). If the selected neighborhood (or village) has more than 1500 households, it is divided into several units, each containing roughly 750 households, and one of the units is randomly selected and used as a PSU. In each PSU, our mappers constructed a list of households by field trips, from which 120 households were randomly selected. One permanent resident aged 15–69 is then chosen randomly in each household. In each household, all age-eligible (15–69 years) household members (i.e., who had been living there for more than 6 of the previous 12 months) was ordered by their gender and age. One member was then selected for the survey by means of a Kish grid [20]. Following the same sampling strategy, a total of 6581 and 6454 respondents were surveyed in 2018 and 2019, respectively. We pooled the observations from the 2018 data (covering 10 districts/counties, 43 communities/townships, 242 neighborhood/villages) and the 2019 data (covering 10 districts/counties, 45 communities/townships, 124 neighborhood/villages) and obtained a sample of 13,035 respondents (covering 10 districts/counties, 77 communities/townships, 362 neighborhood/villages) in total. All respondents who agreed to participate in the survey signed an informed consent form at the beginning of the survey. Our final sample includes 12,360 respondents (aged 15–69), making up 95% of the full sample, after dropping those with non-response to health outcome variables (*n* = 19), non-response to demographic and socioeconomic status information (*n* = 84), non-responses to smoking-related questions (*n* = 6), and finally, lacking information for community population size (*n* = 566).

### 2.2. Measures

#### 2.2.1. Questionnaire Design and Measure of Health Literacy

The questionnaire was developed based on “Basic Knowledge and Skills of People’s Health Literacy (pilot edition)”, which is also called “66 Tips of Health: Chinese Resident Health Literacy Manual” [21]. The 66 Tips of Health document was designed by experts in public health, health education and promotion, and clinical medicine using the Delphi method [22]. Based on the 66 Tips of Health document, a standardized question bank is constructed. The final questionnaire was compiled by the National Institute of Health Education, National Health Commission of China. The questionnaire kept the same format since 2012 and included similar instruments used in the second NHLS. It is worth noting, to ensure comparability across cities, the same health literacy questionnaire is used nationwide, and questions were selected from the question bank to make surveys comparable across survey years [23]. The reliability and construct validity of the health literacy scale is justified by the National Institute of Health Education department and the instruments used in the 2012 questionnaire has Cronbach’s alpha of 0.931 and Spearman-Brown split-half coefficient 0.808 [23]. In a later study with a different sample, the validity of the national health literacy scale is also validated. The overall Cronbach’s alpha of the (original) 80 items in the 2012 questionnaire was 0.95 and the Spearman–Brown split-half coefficient is 0.94 [24]. 

There are 50 health literacy questions in the 2019 questionnaire, which can be divided into three types: true-or-false; multiple-choice questions (with either one or multiple correct answer(s)); and vignette questions. Correct response to each true-or-false and single-answer question counts one, and correct response to a multiple-answer question counts two towards the total score, with the full score being 66. Questions, depending on their relevance to public health, can be divided into six categories: (1) scientific views of health; (2) infectious disease prevention; (3) chronic disease prevention; (4) safety and first aid; (5) medical care; and (6) health information (for more detailed descriptions of the questions, see [24,25]). We define a respondent as having adequate health literacy if the respondent obtained at least 80% of the full score (i.e., 53). This binary classification using 80% threshold has been used consistently over time since the first national survey. By design, these questions are grouped into one of the three dimensions: (1) knowledge and attitudes (22 items); (2) behavior and lifestyle (16 items); and (3) health-related skills (12 items). Following the 80% threshold, we can define whether a respondent has adequate health literacy on each of the three dimensions.

#### 2.2.2. Community Health Supporting Environment Variables

We collected up-to-date data of health supporting environments (or termed healthy settings interchangeably) in Ningbo. The data records nine types of healthy settings, including healthy worksites, healthy communities, health schools, health canteens, healthy restaurants, healthy trails, healthy huts, healthy streets, and healthy parks. Each type of supporting environment is recorded at the neighborhood (in urban areas) or village (in rural areas) level, which we then aggregate at the community level (or townships in rural areas) and merge with the health literacy data at the individual level. Residents from the same communities, therefore, are assigned the same value for each healthy setting variable. Because our outcome variables represent the health of residents randomly selected from each community, we use only the types of settings that are accessible to all the residents living in the community. We do not use healthy worksites, for example, because such a setting is only accessible to people working within the setting. It is not accessible to all residents in the neighborhood. Thus, for the purpose of this study, we use the information of three types of health supporting environments and create our key health supporting environment variables that vary at community level and we explain as follows:Access to healthy parks. A respondent is defined as having access to healthy parks if the respondent lives in a community with at least one healthy park. A park is classified as a healthy park if it has health elements for residents to carry out fitness exercises and to acquire health-related knowledge and skills [8].Access to healthy trails. A respondent is defined as having access to healthy trails if the respondent lives in a community with at least one healthy trail, i.e., a setting set in the community or public areas that is suitable for walking or jogging. Furthermore, a healthy trail should have a certain length and contains health elements and can provide residents with health-related knowledge and skills in the process of fitness activities [8].Access to healthy huts (stations). A respondent is defined as having access to healthy huts if the respondent lives in a community with at least one healthy hut. A healthy hut is a place that provides self-service health testing services and materials for disseminating health knowledge, which aims to promote early detection of chronic diseases and to help the public to self-manage their health by raising their health awareness [8].

#### 2.2.3. Health Outcomes

We use two variables to measure the health condition of respondents—self-reported health status and self-reported chronic diseases. Respondents were asked to rate their general health as being: (1) excellent, (2) good, (3) fair, (4) poor, or (5) very poor. This self-reported variable tells us the perception of general health a resident has, which is one of the instruments to measure the health-related quality of life [26]. In addition, this five-point Likert scale measure is a robust predictor of mortality and correlates strongly with other objective health indicators [27,28,29]. We then use the responses to create a dichotomous measure, which is coded 1 if the response was good/excellent and 0 if the response was fair/poor/very poor. Respondents were asked whether they had any chronic disease and the name of the disease from a list of six diseases, including hypertension, heart problems, cerebrovascular diseases, diabetes, malignant tumor (cancer) and the other.

#### 2.2.4. Control Variables

Our control variables are divided into two groups. Individual level covariates include gender, age, education level, job status, number of household members, household annual income per capita and smoking status. Community level covariates include classification of rural/urban areas, dummies indicating 10 districts/counties and the population size at community level. Age is categorized into three groups as ‘15–44; 45–59; 60–69′. Education level was categorized into four levels: illiterate (used as the base), elementary/middle school, high school, and college or above. Job status was categorized into five groups: working in public sectors (including those whose responses are civil servants, medical workers, teachers, other state enterprises, or students), farmers, manual laborers, working in private sectors, and the other. Household annual income per capita is CPI (consumer price index) adjusted. According to [30], 2019 CPI is 102.9 (preceding year = 100) and we use this figure to convert all 2018 incomes to 2019 price. Respondents who were smoking at the time of the survey were classified as current smokers. Those who smoked before but were not smoking at the time of the survey were classified as former smokers.

### 2.3. Statistical Analyses

Our primary health outcome variable is dichotomous, indicating whether a respondent self-reported good health. We use chronic disease occurrence as an alternative outcome. Key variables of interest are access to healthy parks, access to healthy trails, access to healthy huts (at the community level) and health literacy (at the individual level). Covariates include community level characteristics and demographic and socio-economic status factors of individuals mentioned earlier. To ease interpretation, a linear probability model (LPM) is used to estimate the relationship between community health supporting environment and individual health outcomes. All statistical analyses are performed using StataCorp LLC’s Stata Statistics Software version 15.0 [31].

## 3. Results

### 3.1. Descriptive Statistics

The summary statistics of our variables are presented in Table 1. We report four statistics for the full sample in the first four columns: mean, standard deviation, minimum and maximum value for each variable. Our 12,360 respondents are evenly split between 2018 and 2019 surveys. About 21% respondents are identified as having adequate health literacy (HL), meaning out of 100 adult residents in Ningbo, 21 obtained 80% of the total score from 50 health literacy questions. This rate is above the 2019 national rate of 19.2% [32]. In terms of the health outcomes, about two thirds self-reported having good health and 27% reported having at least one chronic disease.

About 23% of respondents have access to healthy parks in the community where they live, 43% have access to healthy trails and 25% have access to healthy huts. In terms of community level characteristics, about 58% of respondents are living in urban areas, which is in line with the national urbanization rate in 2018 and 2019 [33]. The community population size that a resident lives is 60,000 residents on average.

In terms of individual level characteristics, less than half of our residents are males, which by male-to-female sex ratio is 100:106. An average respondent is aged 50, married (84%), lives together with two other members (average household size is 2.8). The annual household income is on average 37,911 yuan per capita after adjusting for household size. This figure is higher than the 2019 national average of 30,733 yuan [34]. As mentioned earlier, Ningbo city is a port city as well as a commercial and financial center in South China. Thus, on average, our respondents have better socioeconomic status than the national average. In addition, Ningbo is the second biggest city in Zhejiang province. The 2018 GDP per capita for Zhejiang province ranks fifth among 31 provinces in China [35]. In terms of education level, about 8% of the respondents have no education, 56% just finished elementary school or middle, 17% finished high school, and the remaining 20% have a college or higher degree. In terms of occupation, about 12% are working in public sectors (including civil servants, medical workers, teachers, and students), 27% are farmers, 19% are manual laborers, and 21% are working in private sectors. The last ‘other’ group (22%) includes those who are unemployed, retired or not working. Lastly, 22% of our sample are current smokers, 7% are former smokers and the remaining 71% never smoked.

We also split our sample by their level of HL and report the respective mean statistic in the last three columns along with the p-value that indicates the equality of the mean statistics between the two groups. These statistics help us identify the key characteristics that are associated with a higher level of HL. Not surprisingly, respondents with “adequate HL” on average have better health. They are more likely to self-report having good health and they are significantly less likely to have any chronic diseases. In addition, the proportion of respondents with access to the three types of health supporting environments at communities is also greater among the group with adequate HL. Compared to those with “inadequate HL”, those with “adequate HL” are on average more likely to live in urban areas (more populated), younger (43 vs. 51), have a larger household size, and are richer in terms of annual household income per capita. In addition, they are better educated and more likely to work in public sectors or in private sectors). Health literate people are also less likely to smoke, either as current smokers or former smokers.

### 3.2. Baseline Results

We examine the association between health supporting environments and residents’ self-reported health by estimating the probability of self-reported good health in a linear probability model and the results are presented in Table 2. In column (1) we include no covariate but three binary variables indicating access to healthy parks, access to healthy trails and access to healthy huts, respectively, in addition to the binary HL status variable. In column (2) we add community level covariates, including rural/urban classification, survey year and dummies indicating 10 districts/counties. In column (3) we exclude community level covariates and add individual level covariates. In the last column, column (4), we add both community level covariates and individual level covariates. This ordering provides a means to observe how each group of covariates could explain the effects of healthy setting variables and HL variable in column (1).

In our crude specification (column 1), both access to healthy parks and access to healthy trails are significant, while access to healthy huts is insignificant. Access to healthy parks and access to healthy trails are associated with a higher probability of self-reporting good health by 3.2 percentage points and 4.7 percentage points, respectively. In addition, adequate HL is associated with a higher probability of self-reporting good health by 6.2 percentage points. In column (2), the coefficient of access to healthy huts becomes significant with the inclusion of the community level covariates, which suggests respondents living in communities with healthy huts are significantly less likely to self-report good health than their counterparts by 5.4 percentage points. The coefficients on the other three variables are overall unchanged in terms of sign and significance. In column (3) we add individual level covariates. Comparing to column (1), the coefficient on adequate HL diminishes and becomes insignificant. In column (4), with the inclusion of the full covariates, the coefficient on access to healthy parks also diminishes in size but remains significant at 10 percent level.

### 3.3. Interaction Effect of Access to Parks with Health Literacy

In this section, we investigate the extent to which the beneficial effect of access to parks is affected by the HL level of the respondents. Although having adequate HL is insignificant in our baseline regression with full covariates, we now include the interaction term between access to healthy parks and the HL status variable. The results are reported in Table 3.

For comparison, in column (1) we repeat our baseline estimates from Table 2. With the inclusion of the interaction term, the coefficient on access to healthy parks is 3.9 percentage points—higher than the baseline coefficient in column (1). The interaction term is negative and is significant at 1 percent level, suggesting the effect of access to healthy parks is significantly smaller among people with adequate HL by 8 percentage points. For illustration, we calculate the adjusted probabilities of self-reporting good health, with 95% CI (confidence interval) for four distinct groups of residents by their level of HL and availability of healthy parks in their communities. The results are plotted in Figure A1 in the Appendix B. It appears that access to parks has a beneficial effect, but only among those whose HL level is low. In contrast, for a respondent with adequate HL, the beneficial effect of access to parks is at most negligible (the 95% CI overlaps in the last two bars).

Although we follow the national 80% threshold to distinguish adequate and inadequate HL, this classification could be arbitrary. In column (3) in Table 3, we replace the binary HL status variable with the original continuous HL score variable, which ranges from 0 to 66. Like column (2), the interaction term is significant (at 5 percent level). To better illustrate the interaction effect between HL score and access to parks, we calculate the marginal effect of access to healthy parks at different levels of HL score (at an incremental increase of five). The results are plotted in Figure A2 in the Appendix B. As the HL score increases, the marginal effect of access to healthy parks decreases. When the HL score is lower than or equal to 38 (dashed line), the coefficients and 95% CI of access to parks are above zero. When the HL score is above 38, the marginal effect is not significantly different from zero. The 38 cut-off is lower than the median of HL score (i.e., 42) and below the threshold defining adequate HL status (i.e., 53).

### 3.4. Interaction Effect of Access to Healthy Huts with Individual’s Health Literacy

In Table 2 we identified a negative association between access to healthy huts and self-reported good health. Similarly, we investigate the extent to which the seemingly adverse effect of healthy huts relates to the HL level of the residents in Table 4 by including the interaction term between access to healthy huts and HL.

With the inclusion of interaction term in column (2), the size of the coefficient on access to healthy huts increases compared to the baseline coefficient reported in column (1) (from −0.050 to −0.070), while the coefficient on adequate HL status becomes negative (0.039) and is significant at 1 percent level. The coefficient on the interaction term is positive and significant at 1 percent level, suggesting the effect of healthy huts is greater among those with adequate HL than those with inadequate HL by 9 percentage points. We can also interpret the positive interaction effect as the effect of HL is greater among those with access to healthy huts than those without access to healthy huts.

For a better illustration, we calculate the adjusted probabilities of self-reporting good health (with 95% CI) for four groups of residents by their HL status and access to healthy huts. The results are plotted in Figure A3 in the Appendix B. The first two bars show, conditioning on no access to healthy huts, the probability of self-reporting good health is higher among those with inadequate HL (the adjusted probability is 0.69) than that among those with adequate HL (the adjusted probability is 0.65). The difference is significant, as can be read by the nonoverlapping 95% CI. The last two bars show, conditioning on having access to healthy huts, the probability of self-reporting good health is lower among those with inadequate HL (the adjusted probability is 0.62) than that among those with adequate HL (the adjusted probability is 0.67). This difference, however, is not significantly different from zero.

In column (3) in Table 4, we replace the binary HL status variable with the original continuous HL score. The coefficient of access to healthy huts is negative (−0.090), indicating its effect conditioning on the healthy literacy score being zero. Although the interaction term and the HL score are not individually significant, together with the healthy huts variable, they are jointly significant at 5 percent level. As the HL score increases, the marginal effect of access to healthy huts increases (see Figure A4 in the Appendix B). When the level of HL is lower than or equal to 56 (dashed line), the coefficients and 95% CI of access to healthy huts is below zero. When the HL score is above 56, the 95% CI become insignificant. This cut-off of 56 is above the median of HL score (i.e., 42) and the threshold defining adequate HL status (i.e., 53).

### 3.5. Alternative Health Outcome

In this section we examine the health supporting environment effects with a different health outcome—chronic disease occurrence. In a LPM model with full covariates, access to healthy parks is significantly associated with a reduced likelihood of having chronic diseases by 2.7 percentage points (see Table A1 in the Appendix A). The other two types of settings are not significant, we thus only examine the interaction effect between the access to healthy parks and the HL variable. These results are reported in Table 5. The interaction term is positive but insignificant. The access to healthy parks is associated with a reduced likelihood of having chronic diseases, but only among those with a low level of HL. We also plot the effect of access to healthy parks that changes with HL score in Figure A5 in the Appendix B. Like the results we observe for the self-reported health status outcome, access to healthy parks is associated with a better health outcome, but as HL increases, the protective effect diminishes.

## 4. Discussion

We use a sample of 12,360 adults (aged 15–69) surveyed in 2018 and 2019 NHLS survey in Ningbo, China and examine the effects that access to health supporting environments have on residents’ health and well-being. We study the role of HL in moderating the effect of community health supporting environment. By health supporting environments, we pick up three types of health supporting environments at neighborhood level, namely, healthy parks, healthy trails, and healthy huts, and construct the corresponding measures by aggregating at community level, which we then merge with individual level data characterizing residents’ HL, self-reported health status and chronic disease status.

On average, 66% of residents rated their health as being good or excellent as opposed to fair, poor or very poor. We examine the association of self-reported health with healthy parks, healthy trails, and healthy huts in Table 2. We find access to healthy parks or healthy trails is positively associated with residents’ self-reported health controlling for community level covariates and residents’ individual level demographic and socioeconomic characteristics. Parks and trails can improve health in several ways including increased physical activity and improved mental health [36]. For example, walkable access to sites like parks may motive people to participate in physical activity and to do so more frequently. Outdoor activities in parks can also help for stress reduction. In addition, parks can reduce air and water pollution, protect hazard areas from inappropriate development and mitigate urban heat islands [36]. In contrast, we find access to healthy huts is negatively associated with residents’ likelihood of self-reporting good health. It is not unreasonable that on average the proportion of residents self-reporting good health is lower among communities with healthy huts. The testing services in healthy huts enable residents to test their blood pressure and glucose level and they are likely to be alert to their health problems and be encouraged to see doctors. Residents might be diagnosed with diseases if they had not used the services in healthy huts. As a result, they self-report a lower level of health. In the long run, however, early detection prevents the disease to progress to more serious stage. Alternatively, residents might not be diagnosed with any diseases from using the services in healthy huts, but they are likely to evaluate their own health with a more objective perception. Therefore, when being asked to rate their health in general, their responses come with a smaller self-reporting bias. A prior, we do not know whether this bias is positive or negative, but a positive bias is more likely among some groups than others. For example, some smokers may justify or rationalize their smoking behaviors by believing they are healthy [37,38]. For both reasons above, we are likely to observe a lower level of self-reported health among communities with healthy huts.

The above results have some noteworthy implications. Healthy huts might play a role in reducing the likelihood of undiagnosed diseases and changing people’s lifestyle. Disease prevention is a challenge in many countries. Diabetes among adults (18+) in China in 2010 was 12 percent, and pre-diabetes was estimated to be 50 percent with the majority of patients undiagnosed and untreated [39]. If healthy huts do encourage residents to seek care who may otherwise forgo health care due to their perception on their own health, it implies the cases of undiagnosed diseases would be less prevalent among communities with healthy huts. In [37], we find that smokers whose self-reported health is higher are less likely to consider quitting. This relationship persists after we control for their objective health status. In other words, one’s perception of health matters. We speculate healthy huts could serve a role in motivating smokers to quit because smokers perceive their health differently. It is like the effect of being advised to quit by health care professionals.

We investigated the role of HL in Table 3 and Table 4. We find HL plays a role in moderating the effects of healthy parks. The positive association between access to healthy parks and self-reported health is more likely to be observed among people with a low level of HL, which is shown in Table 3 (and illustrated in Figure A1 and Figure A2 in the Appendix B). This might arise because residents with adequate HL have different sources of learning health news. Education is one of the key factors that determine the level of HL, which also plays a key role in explaining how people learn and internalize health information [40]. For example, in one study, it is documented that people with a college education are more likely to receive their most useful health information from books, newspapers, or magazines. In contrast, the less educated (i.e., those with a high school degree or less) are most likely to get their information from a doctor [40]. This difference might imply walkable access to sites, such as parks and trails, may not serve as the primary venue for physical activity (a key ingredient to keep a good health) when people are equipped with better health knowledge. As a result, the effect of healthy parks is less salient among the group with better HL. We find similar results when we replace the health outcome with chronic disease occurrence in Table 5, and we observe a similar pattern in Figure A5 in the Appendix B that access to community healthy parks is associated with a reduced likelihood of having chronic diseases, but only among those with a low level of HL.

The role of HL is different when it comes to the effects of healthy huts. It appears that access to healthy huts is negatively associated with people’s self-reported health status, but only among those with inadequate HL. This might arise because people with adequate HL are more likely to use the services in healthy huts. We lack the data to test this hypothesis as HL of the respondents are not measured before and after their access to healthy huts. In addition, most services in healthy huts are self-service, it is not impossible residents with inadequate HL might misuse or misinterpret the testing results without some prior knowledge. On the other hand, we find the effect of HL is significantly greater among residents with healthy huts. This might arise because HL could improve after the diagnosis of chronic diseases [25]. There is evidence that a negative health event can prompt individuals to adopt risk-reducing behaviors [41,42]. Similarly, the diagnosis of chronic disease might expose individuals to more disease-related information. This improved health knowledge might help them prevent a new disease or promote their health and well-being. In other words, HL is a condition that residents benefit from healthy huts.

It is worth noting, one of the aims of building healthy huts is to facilitate the dissemination of health knowledge and improve individuals’ ability to make health-related decisions [8]. In other words, healthy huts are built to improve residents’ HL. This improved HL may in turn help their use of the services and adopt a healthy lifestyle. If we assume there are two channels through which HL can be acquired—passive pathway through diagnosis of diseases and active pathway through health education, we believe healthy huts promote the health of residents by helping residents to gain health knowledge via an active pathway. Indeed, we find in our descriptive statistics that residents living in communities with healthy huts on average have a higher level of HL. However, this difference disappears after we control for community level and individual level covariates (see Table A2 in the Appendix A).

As a robustness check, we examine the extent to which the health supporting environments effects vary with the gender and age of the residents (see Table A3 in the Appendix A). It appears the male and young samples are more likely to experience the positive effect of access to healthy parks. The positive effect of access to healthy trails does not differ by gender but is only significant in the old sample. The negative effect of healthy huts varies by gender and age of the residents and is more likely to be pronounced in females and the old sample. In contrast, the effect of access to healthy huts is positive in the young sample. In light of our earlier observation in Table 1, it might be because younger residents on average have a higher level of HL, and thus, are more likely to benefit from using healthy huts. We also study the interaction effect between access to healthy parks and HL score and the results are illustrated in Figure A6 and Figure A7 in the Appendix B. 

Although the rural–urban difference in health outcomes and HL is evident in other studies and is also observed in our data (see Table A4 in the Appendix A for the summary statistics of the rural and urban sub-samples), we do not think it is appropriate to examine the effects of health supporting environments separately in the rural and urban data. As mentioned earlier, the national geographic distribution of health supporting environments is not equal and there are more accredited constructions in (more developed) eastern provinces than in (less developed) middle/western provinces. This pattern is also true between rural and urban regions in our data. In Table A4, we find the urban residents on average have a higher level of HL as well as better health than their rural counterparts. The fraction of access to healthy parks, healthy trails, and healthy huts is also significantly greater in the urban sample. For example, over 40% of urban residents are living in communities with healthy huts. In contrast, only 3% of rural residents have access to healthy huts. The estimates would be misleading if we restrict to the rural/urban sub-samples, because the rural-urban split is a key factor that determines the distribution of health supporting environments. Despite this, we report the estimates of this exercise for illustration purpose in Table A5. For ease of comparison, in column (1) we repeat our estimates for the full sample from Table 2. Here, the effect in the full sample (e.g., 0.020 for the access to healthy parks) is hardly a weighted average of the effects in the two subsamples (i.e., −0.064 for the rural sample and 0.013 for the urban sample), an indication that it is inappropriate to use the partitioned data here. However, it does not mean reducing the rural-urban disparity in health supporting environments is not important. With the growth in urbanization, we do not think the growth in health supporting environments would be an independent process with reducing the rural-urban disparity in health outcomes.

Another problem is we use community to define residents’ access to health supporting environments. We assume settings are available to all residents living in the same community, who also have equal likelihood to use the settings. This might not apply to all residents, whose willingness to use the settings cannot be recorded by our data. Theoretically, residents living beyond a community can also access the settings, but due to the geographical proximity, the associated cost should be greater (e.g., transportation cost). Although we do not think it is reasonable to define access to healthy setting at a higher-level geographic entity involving only 10 districts/counties, we repeat our estimations using district level health supporting environments. We find none of three settings are significant except for the access to healthy trails (a positive effect). Furthermore, we consider the level of health supporting environment using binary classification, discarding the information that some communities have more than one healthy setting. Our main findings are unchanged if we use the number instead of the presence (coded 0/1) of health supporting environments. With the trend in the growth of health supporting environments, we consider it a potential area for future research to use geographics data to classify the level of health supporting environments at individual level (i.e., an individual is assigned a unique value based on the size of the environments proportional to the meter circular buffer of their residential address predefined) so to better evaluate and quantify the health supporting environments effects.

There are several limitations in our study, and our findings should be interpreted with caution. Firstly, this is a cross-sectional study. There might exist factors that influence a resident’s health and the accessibility of health supporting environments. For example, residents might choose to move to neighborhoods with better healthy supporting environments for the sake of their health. In addition, although we have controlled a series of community level characteristics (e.g., resident population and district/county dummies), it is likely the location of health supporting environments is correlated with some community characteristics that we do not control for but determine the residents’ health and well-being at the same time, which is likely to undermine the effects of healthy parks we observe. Secondly, our data are not representative nationally, and Ningbo represents one of the cities with better economic development in China. We do not think our findings apply to all regions in China. Lastly, our HL measurement can be limited in measuring HL which is an evolving concept [43] and is not comparable with measures used in other countries. Despite their reliability and construct validity we mentioned earlier, the construct instruments included in the questionnaire are basically unchanged in the past few years. It implies respondents with higher score of HL may not be more knowledgeable, but simply better at taking tests than those who achieved lower scores. 

## 5. Conclusions

We study the association between the accessibility to health supporting environments at community level with residents’ health using 2018 and 2019 NHLS data from Ningbo, a city located on the Yangtze River Delta in South China. We find access to healthy parks and healthy trails—two settings that incorporate health elements in natural environment—are associated with a higher probability of self-reporting good health, which suggests the development of health supporting environments might mitigate the health problems that are associated with urbanization. Residents’ level of HL also has a role to play. Although residents with adequate HL do not obtain additional benefits from accessing healthy parks or healthy trails, they are more likely to benefit from using healthy huts—a setting that enables residents to use self-service to test their blood pressures and glucose levels. The policy implications of our study are twofold. Firstly, residents living in neighborhoods with poor health should be targeted when building health supporting environments. For groups of people with low HL, for example, health supporting environments might be effective in improving their health and well-being. Secondly, considering the negative association between healthy huts and residents’ self-reported health, it is necessary to measure the long-term effect of health supporting environments because the health effects of some settings (e.g., to reduce the undiagnosed diseases and to reduce the positive bias in health perception) are unlikely to be observed in the short-term. Lastly, we believe our study provides an opportunity for trans-disciplinary research at the basis of Sustainability Science, and the data could be analyzed in relation to the physical space and the urban scale, enlarging the multidisciplinary approach of the research.

## Figures and Tables

**Table 1 ijerph-18-07769-t001:** Summary statistics.

	All				Adequate HL		
					No	Yes	
	*Mean*	*S.D.*	*Min*	*Max*	*Mean*	*Mean*	*p*-Value
2019	0.484	0.500	0	1	0.481	0.493	0.308
Adequate health literacy (HL)	0.208	0.406	0	1	0.000	1.000	0.000
Health outcomes							
Self-reported good health	0.666	0.472	0	1	0.652	0.722	0.000
Has any chronic disease	0.273	0.445	0	1	0.301	0.165	0.000
Healthy settings at communities							
Access to healthy parks	0.228	0.420	0	1	0.209	0.301	0.000
Access to healthy trails	0.428	0.495	0	1	0.408	0.505	0.000
Access to healthy huts	0.253	0.435	0	1	0.243	0.290	0.000
Community level characteristics							
Urban	0.576	0.494	0	1	0.545	0.696	0.000
Population (1000)	59.990	34.680	12	181	57.993	67.606	0.000
Individual level characteristics							
Male	0.485	0.500	0	1	0.486	0.481	0.665
Age in years	49.505	13.016	15	69	51.246	42.868	0.000
Aged 15–44	0.328	0.469	0	1	0.266	0.563	0.000
Aged 45–59	0.398	0.490	0	1	0.421	0.312	0.000
Aged 60–69	0.274	0.446	0	1	0.313	0.125	0.000
Married	0.839	0.368	0	1	0.842	0.824	0.028
Household size	2.850	1.261	1	10	2.813	2.989	0.000
HH annual income pc (1000 Yuan)	37.911	52.981	0	2572	34.746	49.981	0.000
Education							
Illiterate	0.079	0.269	0	1	0.097	0.009	0.000
Elementary/Middle	0.557	0.497	0	1	0.617	0.329	0.000
High school	0.165	0.371	0	1	0.149	0.227	0.000
College or above	0.199	0.399	0	1	0.138	0.434	0.000
Job status							
Public sectors	0.116	0.320	0	1	0.088	0.220	0.000
Farmers	0.269	0.444	0	1	0.312	0.107	0.000
Manual laborers	0.188	0.390	0	1	0.193	0.167	0.002
Private sectors	0.209	0.407	0	1	0.182	0.313	0.000
Other	0.218	0.413	0	1	0.225	0.193	0.001
Smoking status							
Never	0.708	0.455	0	1	0.690	0.775	0.000
Quit	0.074	0.262	0	1	0.081	0.049	0.000
Now smoke	0.218	0.413	0	1	0.229	0.176	0.000

Source: 2018, 2019 National Health Literacy Surveillance (NHLS) survey in Ningbo. Notes: *N* = 12,360. HH annual income pc is CPI adjusted in 2019 price.

**Table 2 ijerph-18-07769-t002:** OLS estimates on self-reported health: effects of health settings and health literacy.

Variables	(1)	(2)	(3)	(4)
	Coef	Coef	Coef	Coef
Access to healthy parks	0.032 ***	0.025 **	0.027 ***	0.020 *
	(0.010)	(0.012)	(0.010)	(0.012)
Access to healthy trails	0.047 ***	0.058 ***	0.049 ***	0.060 ***
	(0.009)	(0.011)	(0.009)	(0.011)
Access to healthy huts	0.006	−0.054 ***	−0.014	−0.050 ***
	(0.010)	(0.016)	(0.010)	(0.015)
Adequate HL	0.062 ***	0.053 ***	−0.013	−0.013
	(0.010)	(0.010)	(0.011)	(0.011)
Comm. covariates	No	Yes	No	Yes
Indi. covariates	No	No	Yes	Yes
Dep Mean	0.666	0.666	0.666	0.666
Adj. R^2^	0.007	0.019	0.051	0.058
Obs	12,360	12,360	12,360	12,360

Notes: (1) dependent variable = 1 if a resident self-reported good health, 0 otherwise. (2) Comm. covariates, community level covariates, include classification of rural/urban areas, dummies indicating 10 districts/counties and the population size at community level. (3) Indi. covariates, individual level covariates, include gender, age, education level, job status, number of household members, household annual income per capita and smoking status. (4) Standard errors in parentheses * *p* < 0.1, ** *p* < 0.05, *** *p* < 0.01.

**Table 3 ijerph-18-07769-t003:** OLS estimates on self-reported health: interaction effects of healthy parks with health literacy.

Variables	(1)	(2)	(3)
	Coef	Coef	Coef
Access to healthy parks	0.020 *	0.039 ***	0.085 ***
	(0.012)	(0.013)	(0.033)
Adequate HL	−0.013	0.009	
	(0.011)	(0.012)	
Access to health parks × Adequate HL		−0.080 ***	
		(0.022)	
HL total score			−0.000
			(0.000)
Access to healthy parks × HL total score			−0.002 **
			(0.001)
Comm. covariates	Yes	Yes	Yes
Indi. covariates	Yes	Yes	Yes
Dep Mean	0.666	0.666	0.666
Adj. R^2^	0.058	0.059	0.059
Obs	12,360	12,360	12,360

Notes: (1) dependent variable = 1 if a resident self-reported good health, 0 otherwise. (2) HL total score, health literacy total score. (3) Comm. covariates, community level covariates, include classification of rural/urban areas, dummies indicating 10 districts/counties and the population size at community level. (4) Indi. covariates, individual level covariates, include gender, age, education level, job status, number of household members, household annual income per capita and smoking status. (5) Standard errors in parentheses * *p* < 0.1, ** *p* < 0.05, *** *p* < 0.01.

**Table 4 ijerph-18-07769-t004:** OLS estimates on self-reported health: interaction effects of healthy huts with health literacy.

Variables	(1)	(2)	(3)
	Coef	Coef	Coef
Access to healthy huts	−0.050 ***	−0.070 ***	−0.090 ***
	(0.015)	(0.016)	(0.034)
Adequate HL	−0.013	−0.039 ***	
	(0.011)	(0.013)	
Access to healthy huts × Adequate HL		0.091 ***	
		(0.022)	
HL total score			−0.001 *
			(0.000)
Access to healthy huts × HL total score			0.001
			(0.001)
Comm. covariates	Yes	Yes	Yes
Indi. covariates	Yes	Yes	Yes
Dep Mean	0.666	0.666	0.666
Adj. R^2^	0.058	0.059	0.058
Obs	12,360	12,360	12,360

Notes: (1) dependent variable = 1 if a resident self-reported good health, 0 otherwise. (2) HL total score, health literacy total score. (3) Comm. covariates, community level covariates, include classification of rural/urban areas, dummies indicating 10 districts/counties and the population size at community level. (4) Indi. covariates, individual level covariates, include gender, age, education level, job status, number of household members, household annual income per capita and smoking status. (5) Standard errors in parentheses * *p* < 0.1, *** *p* < 0.01.

**Table 5 ijerph-18-07769-t005:** OLS estimates on chronic disease incidence: interaction effects of healthy parks with health literacy.

Variables	(1)	(2)	(3)
	Coef	Coef	Coef
Access to healthy parks	−0.027 ***	−0.031 ***	−0.064 **
	(0.010)	(0.012)	(0.030)
Adequate HL	−0.012	−0.016	
	(0.009)	(0.010)	
Access to healthy parks × Adequate HL		0.014	
		(0.018)	
HL total score			−0.000
			(0.000)
Access to healthy parks × HL total score			0.001
			(0.001)
Comm. covariates	Yes	Yes	Yes
Indi. covariates	Yes	Yes	Yes
Dep Mean	0.273	0.273	0.273
Adj. R^2^	0.170	0.170	0.170
Obs	12,360	12,360	12,360

Notes: (1) dependent variable = 1 if a resident has at least one chronic disease, 0 otherwise. (2) HL total score, health literacy total score. (3) Comm. covariates, community level covariates, include classification of rural/urban areas, dummies indicating 10 districts/counties and the population size at community level. (4) Indi. covariates, individual level covariates, include gender, age, education level, job status, number of household members, household annual income per capita and smoking status. (5) Standard errors in parentheses ** *p* < 0.05, *** *p* < 0.01.

## Data Availability

The datasets used and analyzed in the current study are not publicly available.

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
