# Peer review of "The Community Health Supporting Environments and Residents’ Health and Well-Being: The Role of Health Literacy"

_ijerph, 2021, doi:10.3390/ijerph18157769_

Round 1

Reviewer 1 Report

I read with interest the paper, I would suggest some changes that can improve the manuscript.

  1. Introduction

I would focus the topic of the paper also in the context of Sustainable Development and the UNESCO Sustainable Development Goals.

Furthermore, the concepts of Wellbeing in the field of urban spaces and architecture and Psychology of Sustainability of Sustainable Development could be introduced.

  1. Materials and Methods

In order to better understand the case study, a map of the cities with geographical references and the location of the areas could be inserted.

  1. Discussion

It would be interesting to investigate the spatial relationship between healthy parks, healthy trails and healthy huts (meaning their location and distribution within the urban tissue), and the distribution of the resident population and samples within the city.

  1. Conclusion

Possible future research developments could be added and be examinated in depth. In this perspective, the data could be analysed in relation to the physical space and the urban scale enlarging the multidisciplinary approach of the research.

Reviewer 2 Report

File attached.

Round 2

Reviewer 2 Report

Thank you for revising the manuscript in accordance with the comments. The impact of health literacy on wellness and environments have now been addressed.